# Evaluation of the Harvest Dates for Three Major Cultivars of Blue Honeysuckle (*Lonicera caerulea* L.) in China

**DOI:** 10.3390/plants12213758

**Published:** 2023-11-02

**Authors:** Min Yu, Songlin Li, Ying Zhan, Zhiqiang Huang, Jinjiao Lv, Yu Liu, Xin Quan, Jinyu Xiong, Dong Qin, Junwei Huo, Chenqiao Zhu

**Affiliations:** 1College of Life Science, Northeast Agricultural University, Harbin 150030, China; 2National-Local Joint Engineering Research Center for Development and Utilization of Small Fruits in Cold Regions, National Development and Reform Commission, Harbin 150030, China; 3College of Horticulture & Landscape Architecture, Northeast Agricultural University, Harbin 150030, China; 4Institute of Agricultural Science and Technology, Bureau of Agriculture and Rural Affairs of Xiaochang County, Xiaogan 100125, China; 5Key Laboratory of Biology and Genetic Improvement of Horticultural Crops (Northeast Region), Ministry of Agriculture and Rural Affairs, Harbin 150030, China

**Keywords:** honeyberry, fruit quality, harvest time evaluation

## Abstract

Blue honeysuckle (*Lonicera caerulea* L.) is an emerging fruit crop; however, determining its proper harvest time in commercial cultivation remains challenging due to its rapid fruit development characteristics. In this study, we investigated 17 agronomic traits of three blue honeysuckle cultivars harvested on 5 successive dates within their respective harvest windows. ‘Lanjingling’, ‘Wulan’, and ‘Berel’ showed solid–acid ratios (SS:TA) ranging from 10.00 to 16.01, 8.13 to 10.23, and 5.77 to 7.11, respectively; anthocyanin contents ranged from 233.85 to 276.83 mg/100 g, 236.38 to 312.23 mg/100 g, and 235.71 to 334.98 mg/100 g, respectively; vitamin C contents ranged from 88.43 to 99.68 mg/100 g, 108.13 to 191.23 mg/100 g, and 89.71 to 120.40 mg/100 g, respectively; phenolic contents ranged from 25.22 to 37.59 mg/g, 25.40 to 36.52 mg/g, and 37.66 to 50.00 mg/g, respectively. The results revealed the SS:TA value consistently increased with delayed harvesting and were significantly negatively correlated with fruit firmness, total acidity, shelf life, and respiration intensity, suggesting it is an ideal maturity indicator for blue honeysuckle berries. The factor analysis suggests that the suitable harvest date for ‘Lanjingling’ could be either 47 days after flowering (DAF) with an SS:TA value of approximately 10.0, characterized by high firmness, extended shelf life, and elevated levels of anthocyanins and phenolics; or 67 DAF (SS:TA ≈ 16.0), characterized by high vitamin C content and sweetness, and larger size and weight. For ‘Wulan’, it suggests the suitable harvest date is either 54 DAF (SS:TA ≈ 9.0), yielding fruit with high levels of anthocyanins and vitamin C; or 62 DAF (SS:TA > 10.0), yielding fruit with high sweetness and large size and weight. For ‘Berel’, it is suggested to be either 52 DAF (SS:TA ≈ 6.5), resulting in fruit with high levels of anthocyanins and vitamin C; or 62 DAF (SS:TA > 7.0), resulting in balanced levels of the fruit quality traits.

## 1. Introduction

Blue honeysuckle (*Lonicera caerulea* L.) is a perennial deciduous shrub belonging to the Caprifoliaceae family. The fruit of the blue honeysuckle is a small berry, ranging in color from azure-blue to black-blue, with a sour to sweet taste and commonly known as ‘honeyberry’, ‘haskap’ or ‘honeysuckle berry’. Traditionally, blue honeysuckle berries have been wild-harvested in Russia, China, Korea, and Japan as an ethnomedicine [1]. Over the past three decades, blue honeysuckle has garnered significant attention from fruit researchers due to its exceptional cold tolerance during dormancy, remarkable frost resistance of buds and flowers in early spring, and early ripening of fruit in summer [2,3,4]. Consequently, a series of elite cultivars have been bred in Eastern Europe [5], northeastern China [4], Hokkaido in Japan [6], and Saskatchewan in Canada [7]. Currently, there has been substantial annual growth observed in the commercial cultivation and processing industry of blue honeysuckle. In 2021, European experts estimated that the global cultivation area of blue honeysuckle had reached approximately 5500 hectares excluding Asian countries [8]; Chinese experts estimated that by 2022, it was nearly 5000 hectares in China alone (mainly concentrated in northeastern provinces) with dominant cultivars including ‘Lanjingling’, ‘Wulan’, and ‘Berel’ [9]. Blue honeysuckle berries are extensively utilized in the production of juice, wine, pastries, jams, candy, and dairy products. In addition, the burgeoning demand for fresh blue honeysuckle berries on the market is further fueled by the rapid advancements in cold-chain logistics [10]. Owing to its elevated concentration of health-promoting phytochemicals such as phenolics, anthocyanins, vitamin C, and iridoids [11,12,13,14], blue honeysuckle exhibits potential applications in functional foods, nutraceuticals, and medicine [15,16]. Previous studies have documented that cyanidin-3-O-glucoside is the predominant anthocyanin in blue honeysuckle berries, with a concentration range of 67.7–886.6 mg/100 g [11,17,18]; the total phenolics of blue honeysuckle berries exhibited a range of 12.5–87.5 mg/g with the majority being chlorogenic acid (20.7–44.0 mg/100 g), neochlorogenic acid (2.0–5.0 mg/100 g), and caffeic acid (0.1–0.2 mg/100 g) [19,20,21]; the vitamin C content ranged from 17.8 to 421.0 mg/100 g [21,22,23]. Consequently, blue honeysuckle has earned its place among ‘emerging fruit crops’, referring to recently domesticated species known for the health-promoting properties of their fruits [24,25]. However, the lack of fundamental and practical agronomic knowledge has significantly impeded its targeted breeding and cultivation standardization, thereby constraining the expansion of its industry.

The determination of the appropriate harvest timing for blue honeysuckle poses a challenge in large-scale commercial cultivation worldwide, primarily due to its successive flowering and fruit setting (lasting approximately one week), relatively short fruit development period (6–9 weeks depending on genotype), rapid fruit ripening, and auto-abscission, as well as its non-climacteric fruit type [22,26,27]. Similar to small berry crops with comparable biological characteristics, such as blueberry, blackcurrant, and strawberry, the fruit quality of blue honeysuckle is significantly influenced by harvest time, in addition to being affected by environmental factors, genotypes, and their interaction [21,28,29,30,31]. Additionally, considering the relatively low firmness and high perishability of blue honeysuckle berries [32], the timing of harvest plays a crucial role in determining fruit quality, which subsequently influences the appropriate harvesting method (manual or mechanized), market destination (fresh market or processing factory), and corresponding postharvest treatment (cold or frozen storage). Therefore, precise assessment of the suitable harvest time for blue honeysuckle not only holds significant practical implications in the field and provides valuable insights for the standardizing cultivation systems but also contributes to advancements in physiology and post-harvest research. However, despite the global recognition of the significance of proper harvest timing for blue honeysuckle, limited attention has been devoted to this crucial agronomic procedure in previous studies [22,26]. A comprehensive evaluation of harvest timing for any cultivars or production regions is currently lacking, even though empirical harvest windows for some cultivars have been documented from west to east [4,26].

This study aims to determine the proper harvest time for the three main blue honeysuckle cultivars (‘Lanjingling’, ‘Wulan’, and ‘Berel’) in China by characterizing and analyzing the dynamic changes in fruit quality traits during their respective harvest windows. To our knowledge, this research represents the pioneering effort toward evaluating the harvesting times for blue honeysuckle berries worldwide. The findings would not only provide a practical approach for large-scale harvesting in commercial cultivation of blue honeysuckle in China but also serve as a valuable reference for its potential global industry expansion.

## 2. Materials and Methods

### 2.1. Plant Materials and Experimental Environment

‘Lanjingling’ (China National Plant Variety Protection: CNPVP-20200389) and ‘Wulan’ (CNPVP-20200390) were bred by Northeast Agricultural University (NEAU), while ‘Berel’ (Heilongjiang Province regional registration of crop varieties: 2011037) is a Russian hybrid cultivar that was introduced to China in 2001 [4]. The experimental bushes, aged six years, were planted with a spacing of 1.0 m × 2.6 m at the Horticultural Experiment Station of NEAU in Harbin, China (E126°43.57′, N45°44.23′). For each harvest date, three replications per cultivar were randomly selected, with each replication consisting of six bushes. The experimental location exhibits a monsoon-influenced humid continental climate (Dwa) according to the Köppen classification [33]. The annual maximum and minimum temperature were recorded at approximately 35 °C and −30 °C respectively, corresponding to zone ‘4a’ in the USDA plant hardiness zones list (https://planthardiness.ars.usda.gov/ (accessed on 1 January 2023)). The meteorological data during the harvest season at the experimental location are presented in Appendix A.

During the 2022 harvest season, the first harvest date for each cultivar was determined to be the date when 95% of the fruit had turned blue. Subsequently, based on the empirical harvest windows for the three cultivars at the experimental location [4], which were ~2 weeks for ‘Wulan’ and ~3 weeks for ‘Lanjingling’ and ‘Berel’, five harvest dates were determined, with four days of interval for ‘Wulan’ and five days of interval for ‘Lanjingling’ and ‘Berel’. Namely, ‘Lanjingling’ and ‘Berel’ were harvested at 47 days after flowering (47DAF), 52DAF, 57DAF, 62DAF and 67DAF; ‘Wulan’ was harvested at 46DAF, 50DAF, 54DAF, 58DAF, and 62DAF. The fruits were hand-harvested, ensuring a minimum of 300 fruits per replication and at least 50 fruits per bush. The harvested fruits were transported to the laboratory using prechilled coolers (Excursion 47L, Esky, Sydney, Australia) and stored in a refrigerator set at 4 °C for 8 h before subsequent measurements.

### 2.2. Fruit Appearance and Physiological Characteristics

For each replicate, 30 sound fruits were randomly selected. The fruit weight was measured using a high-precision electronic balance (QUINTIX124-1S, Sartorius, Göttingen, Germany) with a sensitivity of 0.001 g. The vertical and transverse diameters (h and w) were measured using a digital vernier caliper (DDL313150, Deli, Ningbo, China) with a sensitivity of 0.01 mm. The fruit volume for ‘Lanjingling’ was calculated using the formula ‘π × (w/2)^2^ × h’ based on its cylindrical fruit shape, while for ‘Wulan’ and ‘Berel’ with oval-shaped fruits, the calculation was performed using the formula ‘4/3 π × (w/2)^2^ × h’. The firmness was quantified using a digital fruit durometer (GY-4, HandPi, Yueqing, China) equipped with a 3.5 mm diameter plunger, adhering to the manufacturer’s specifications for one punch per fruit. The fruit color was quantified using a spectrocolorimeter (CM-700d, KONICA MINOLTA, Tokyo, Japan) with three technical replicates. The measurements were performed in the CIELAB color space system (*L***a***b** ) employing a 10° observer and D65 illuminant. The respiration intensity was quantified using a specialized instrument designed specifically for measuring the respiration of fruits and vegetables (SYS-GH30A, Saiyass, Changchun, China); in each experimental replication, a total of 6 fruits were placed into a breathing chamber with a volume of 0.10 L; the measurement was conducted three times at intervals of 30 min. The shelf life was investigated under controlled environments: 4 °C with 90% air humidity (referred to as ‘low-temperature shelf’) and 25 °C with shade (‘referred to as room temperature shelf’); 100 intact fruits for each replication were individually stored in air-permeable plastic containers without any stacking; the fruits were examined every 24 h to determine the count of sound fruits (excluding those showing signs of shriveling, fracture, juicing, or decay); the shelf life (days) was recorded until the percentage of sound fruit dropped below 15%.

### 2.3. Measurements of the Total Soluble Solid, Acid, Phenols, Anthocyanins, and Ascorbic Acid

The fruits were homogenized using a high-speed tissue homogenizer (MR9501, Morphy Richards, Swinton, UK) with 100 fruits per replication, and the resulting homogenate was subsequently filtered to obtain raw juice for the following measurements. The soluble solids (%) and acidity (%) were determined using a handheld refractometer (PaL-BXIACID F5, ATAGO, Tokyo, Japan) following the manufacturer’s specifications. The total anthocyanins were quantified using the pH differential method [34] and expressed as cyanidin-3-galactoside equivalent (mg/100 g·FW) based on the molar absorbance of e = 30,200 and molecular weight of 449.2. The ascorbic acid (vitamin C) content was determined using molybdenum blue colorimetry [35] and is expressed as ‘mg/100 g·FW’. The total phenolics were quantified using an improved Folin–Ciocalteu (FC) method [17] with a microplate reader (Synergy H1, BioTek, Winooski, VT, USA) and the results were expressed as gallic acid equivalent (mg/g·FW). All the detailed extraction and measurement methods are shown in Appendix A.

### 2.4. Statistical Analysis

The raw data obtained from instruments and written records were manually imported into Microsoft Excel 2021 for conducting descriptive statistics (Appendix A). To assess the statistical significance among the five harvest dates of each cultivar (*p* < 0.05 was considered statistically significant), one-way ANOVA was performed using IBM SPSS Statistics for Windows (Version 27.0) [36] with post hoc tests conducted via the Tukey HSD test. The Pearson’s correlation coefficients of the fruit quality traits was calculated and plotted using Chi-plot (https://www.chiplot.online/ (accessed on 15 October 2023)). The principal component bi-plot was constructed using OriginPro 2021 [37] based on 11 fruit quality traits (weight, size, SS, TA, SS:TA, Ff, 4 °C-Sl, 25 °C-SL, anthocyanins, vitamin C, and phenolics). To evaluate the harvest dates for each cultivar, factor analysis was performed using IBM SPSS v27.0 based on nine key quality indicators: weight, size, SS:TA, Ff, 4 °C-Sl, 25 °C-Sl, anthocyanins, vitamin C, and phenolic contents. The parameters were set as ‘descriptive statistics’ of ‘Initial solution’, ‘correlation matrix’ of ‘The Bartlett Test of Sphericity and Kaiser–Meyer–Olkin (KMO)’, ‘extraction method’ of ‘Principal components’; ‘rotation method’ of ‘Varimax rotation’, and factor scores were saved as variables with method of ‘regression’; the other parameters were set as default. The comprehensive factor scores of the 5 harvest dates for each cultivar was computed based on the score of rotation-sums-of-squared-loadings, following the formula: comprehensive factor score = (factor1 score × percentage of variance by factor1 + factor2 score × percentage of variance by factor2 + factor3 score × percentage of variance by factor3)/cumulative percentage of variance by (factor1 + factor2 + factor3).

## 3. Results and Discussion

### 3.1. Fruit Appearances

To assess the changes in fruit appearance of the three cultivars during their respective harvest windows, we investigated fruit size, weight, and color (Figure 1 and Table 1). ‘Lanjingling’, ‘Wulan’, and ‘Berel’, respectively, exhibited fruit length range of 23.08–23.78 mm, 18.30–20.06 mm, and 17.32–17.40 mm, width range of 9.50–9.65 mm, 10.37–10.75 mm, and 11.05–11.10 mm, estimated fruit size range of 1.64–1.73 cm^3^, 2.06–2.42 cm^3^, and 2.21–2.25 cm^3^, and fruit weight range of 1.17–1.19 g, 1.02–1.07 g, and 1.12–1.15 g. The later harvested fruits generally exhibit slightly larger sizes and heavier weights compared to the earlier ones. Statistically significant differences were only observed in ‘Wulan’, respectively between 46DAF and 58DAF, and between 46DAF and 62DAF. Regarding fruit color, the three cultivars exhibited dark grey to dark blue hues in the *L***a***b** coordinate system throughout their harvest windows. The *L** values of ‘Lanjingling’, ‘Wulan’, and ‘Berel’ ranged from 33.99 to 39.45, 28.77 to 37.24, and 28.26 to 34.58, respectively; their *a** values ranged from −1.48 to −1.36, −1.84 to −1.01, and from −1.69 to −0.91, respectively; their *b** values ranged from −5.21 to −6.77, −4.23 to −7.24, and from −3.65 to −5.24, respectively. In addition, for all three cultivars, the absolute values of *L** and *b** reached their peaks at the third or fourth dates and slightly decreased thereafter, indicating a gradual lightening (*L**) and bluing (*b**) of fruit color as harvesting was delayed. The *a** value exhibited irregular changes from the first to the last harvest date.

The appearance properties of fruits play crucial roles in assessing consumer preference and acceptance of blue honeysuckle berries [23]. In this study, there were minimal differences observed in fruit size and weight across the harvest windows, except for 46DAF of ‘Wulan’. These findings suggest, at least for ‘Lanjingling’ and ‘Berel’, the fruit harvested at all five harvest dates can be considered commercially mature, supporting the empirically defined start of harvest (95% of berries have turned blue) and their harvest windows (~3 weeks) in the northeast region of China [26]. The coloration of berries is influenced by a synergistic interplay between cuticular wax and pigment metabolites within the fruit peel [38]. Both the present and the previous findings [22,39,40] suggest that the decrease in the absolute values of *L** and *b** observed at the final harvest date can be attributed to both the degradation of cuticular wax and the dynamic synthesis of pigment metabolites, which are also likely influenced by factors such as wind strength, light intensity, air humidity, temperature, and internal metabolism. Thus, considering the rapid fruit ripening of blue honeysuckle, color indicators may not provide a reliable means for monitoring the ripeness and determining the suitable harvest time.

### 3.2. Fruit Firmness, Respiration Intensity, and Storage Capacity

To assess the tolerance to transportation and storage of the three cultivars harvested at the five different dates, we investigated fruit firmness (Ff), respiration intensity (Ri), and shelf life under 25 °C (25 °C-Sl) and 4 °C (4 °C-Sl) (Table 2). Overall, the fruits harvested at the initial dates exhibited the highest Ff, Ri, 25 °C-Sl, and 4 °C-Sl values, and the values gradually decreased along with the delay in harvesting. The Ff of ‘Lanjingling’, ‘Wulan’, and ‘Berel’ exhibited respective declines of 2.69-fold (3.87N/1.44N), 2.53-fold (2.23N/0.88N), and 2.19-fold (2.94N/1.34N) from the first date to the last date. The most substantial declines in Ff for ‘Lanjingling’, ‘Wulan’, and ‘Berel’ were observed during the periods of 47DAF to 52DAF (from 3.86N to 2.90N), 54DAF to 58DAF (from 1.69N to 1.16N), and 62DAF to 67DAF (from 1.90N to 1.34N), respectively, accounting for approximately one-third of the total decrease. For Ri, consistent decreasing trends were observed throughout the harvest windows of the three cultivars. Notably, ‘Wulan’ consistently exhibited a higher level of Ri (1.5–2 folds) compared to ‘Lanjingling’ or ‘Berel’. Along with the delay in harvesting, the 25 °C-Sl of ‘Lanjingling’, ‘Wulan’, and ‘Berel’ respectively decreased from 12.33 days to 4.33 days, from 11.33 days to 2.33 days, and from 12.00 to 2.33 days. The 25 °C-Sl of the three cultivars exhibited relatively smooth decreasing trends, namely, no significant difference was observed between each two successive harvest dates. The 4 °C-Sl of ‘Lanjingling’, ‘Wulan’, and ‘Berel’ respectively decreased from 26.33 days to 11.00 days, from 21.00 to 5.00 days, and from 24.67 to 12.33 days, along with the delay in harvesting. The 4 °C-Sl of ‘Lanjingling’ showed no significant descent between any two successive dates; that of ‘Wulan’ showed two significant descents (50DAF to 54DAF and 58DAF to 62DAF); that of ‘Berel’ showed one significant descent occurring between 47DAF and 52DAF.

Fruit firmness is a pivotal agronomic trait for small berry crops, influenced by the species’ maturation physiology, and significantly impacts fruit’s shelf life and transportation capacity [41]. The simultaneous decrease of Ff, Ri, and Sl with the delay in harvesting is a prevalent phenomenon observed in berry crops [22,27,42], which can be attributed to alterations in the composition and structure of cell walls induced by maturation and senescence processes [43]. In addition, the respiration intensity of the three cultivars exhibited a consistent decline in our findings, providing partial support for the non-climacteric nature of blue honeysuckle berries [27]. Notably, the constitutive higher Ri of ‘Wulan’ compared to ‘Lanjingling’ and ‘Berel’ might explain its lower Ff and shorter Sl. Moreover, considering the exceptionally high Ri observed in ‘Wulan’ harvested at 46DAF and 50DAF, coupled with the significantly smaller size and lighter weight of ‘Wulan’ harvested at 46DAF (Table 1), these findings imply that the start date of harvesting and the harvest window for ‘Wulan’ should be slightly postponed and shortened respectively. The fruit harvested at 47DAF of ‘Lanjingling’ exhibited an Ff exceeding 3.5N (3.86N), which closely approximates the medium firmness level observed in co-distributed blueberry cultivars [44,45]. Under simulated room temperature (25 °C), fruits harvested during 47DAF to 57DAF of ‘Lanjingling’, 46DAF to 50DAF of ‘Wulan’, and 47DAF to 52DAF of ‘Berel’ exhibited Sl longer than one week. This finding indicates that early harvesting is necessary to reduce postharvest losses for blue honeysuckle berries and suggests the early harvested berries are more suitable for supplying local fresh markets under room temperature conditions and quick-freezing is necessary for preserving the late harvested fruits. Furthermore, it is noteworthy that the shelf life of blue honeysuckle berries was extended by at least twice as long when subjected to 4 °C temperatures compared to 25 °C, indicating the indispensability of rapid precooling and cold storage techniques.

### 3.3. Soluble Solids (SS) and Titratable Acid (TA) Content

The primary fruit tastes were evaluated by analyzing the soluble solids (SS), titratable acidity (TA), and SS:TA ratio (Table 3). Throughout the harvest windows, the SS of ‘Lanjingling’, ‘Wulan’, and ‘Berel’ exhibited ranges of 13.84–15.15%, 16.13–17.65%, and 12.88–16.48%, respectively. The SS of ‘Lanjingling’ exhibited a gradual decline as the harvesting was delayed, resulting in an average value of 14.54%, while the SS of ‘Wulan’ remained relatively stable throughout its harvest window and displayed a high average value of 17.06%. For ‘Berel’, the SS increased until reaching its peak at 57DAF, followed by a gradual decrease towards the lowest value observed at the final harvest date, leading to an average value of 14.09%. In terms of TA, ‘Lanjingling’, ‘Wulan’, and ‘Berel’, respectively, exhibited TA ranges of 0.86–1.52%, 1.85–2.12%, and 1.87–2.51% throughout their respective harvest windows. The TA of ‘Lanjingling’ and ‘Wulan’ showed continuous descending patterns, resulting in average values of 1.16% and 1.92%, respectively, while that of ‘Berel’ displayed a fluctuating trend with the highest value at 57DAF and the lowest value at 67DAF, resulting in an average value of 2.14%. The SS:TA of ‘Lanjingling’, ‘Wulan’, and ‘Berel’ exhibited consistent upward trends along with the delayed harvesting, resulting in ranges of 10.00–16.01 (mean = 12.91), 8.13–10.23 (mean = 8.98), and 5.77–7.11 (mean = 6.62), respectively. Significant increases in SS:TA were only observed during the periods from 47DAF to 52DAF and from 62DAF to 67DAF for ‘Lanjingling’.

The increasing trends of SS and the decreasing trends of TA along with the delay in harvesting have been widely reported in most fruit crops [42,46,47]. In this study, the increases in SS and decreases in TA of the three cultivars exhibited relatively greater stability compared to previous reports [22,26]. The extended duration of fruit development and delayed initiation of harvest in our experimental area may account for this phenomenon, resulting in a more homogeneous fruit ripening compared to regions in Eastern Europe or Canada. Significantly, the SS:TA exhibited a consistent upward trend for all three cultivars as harvesting was delayed, aligning with previous findings [22,26]. This suggests that SS:TA serves not only as an intuitive indicator for assessing sweetness but also potentially functions as a reliable marker for monitoring the maturity of blue honeysuckle berries. Intriguingly, in comparison to previously reported cultivars in Eastern Europe and North America [1,5,7,19,20,21,22,23,39], the three cultivars in present study exhibited moderate to high levels of SS, low to moderate levels of TA, and high SS:TA. These characteristics might represent the regional features of blue honeysuckle berries harvested in Northeast Asia.

### 3.4. Anthocyanins, Phenolics and Vitamin C Contents

To assess the variations in functional constituents of the three cultivars throughout the harvesting windows, we measured the levels of total anthocyanins, phenolics, and vitamin C (Table 4). The total anthocyanins of ‘Lanjingling’, ‘Wulan’ and ‘Berel’ ranged from 233.85 to 276.83 mg/100 g (mean = 253.69 mg/100 g), 236.38 to 312.23 mg/100 g (mean = 273.29 mg/100 g), and 235.71 to 334.98 mg/100 g (mean = 282.01 mg/100 g), respectively; the highest concentrations of anthocyanins were observed at 47DAF for ‘Lanjingling’, at 54DAF for ‘Wulan’, and at 52DAF for ‘Berel’. The total vitamin C of ‘Lanjingling’, ‘Wulan’ and ‘Berel’ ranged from 88.44 to 99.68 mg/100 g (mean = 92.52), 108.13 to 191.23 mg/100 g (mean = 134.92), and 89.71 to 120.40 mg/100 g (mean = 104.49), respectively, with the highest concentrations observed at 67DAF for ‘Lanjingling’, at 54DAF for ‘Wulan’, and at 52DAF for ‘Berel’. The total phenolic of ‘Lanjingling’, ‘Wulan’, and ‘Berel’ ranged from 28.88 to 37.59 mg/100 g (mean = 30.97), 25.40 to 36.52 mg/100 g (mean = 31.14), and 37.66 to 50.00 mg/100 g (mean = 41.52), respectively; the highest concentrations of total phenolics were observed at the first harvest dates (47DAF, 46DAF, and 47DAF).

The bioactive metabolites present in blue honeysuckle berries, primarily anthocyanins, vitamin C, and phenolics, not only serve as essential constituents for plants but also represent the most widely recognized functional components by fruit consumers. Consequently, these compounds play a crucial role in determining the market value of blue honeysuckle berries for applications such as natural pigment production or nutritional supplementation [48,49,50]. The accumulation patterns of the three contents throughout the harvest windows exhibited relatively fluctuating and irregular trends when compared to those of SS, TA, and SS:TA. This phenomenon might be attributed to the distinct microclimatic conditions during the specific harvest dates, as well as the dynamic processes of synthesis and metabolism of these metabolites throughout fruit ripening [51,52], which underscores the potential application of smart equipment for monitoring the fruit ripening of blue honeysuckle and implies the necessity for molecular technology to unravel the synthesis and regulation mechanisms of these three metabolites in blue honeysuckle. Intriguingly, the three cultivars exhibited comparatively lower levels of total anthocyanins and moderate to high levels of total vitamin C and total phenolics in comparison to the European and North American cultivars [1,5,7,19,20,21,22,23,39], thereby probably reflecting distinctive regional characteristics of blue honeysuckle berries harvested in northeastern Asian areas. Moreover, given the primary objective of this study is to offer an approach to farmers, fruit dealers, and processing factory owners for assessing fruit maturity, only viable measurement techniques that do not heavily rely on instruments were employed (including pH-differential analysis, molybdenum blue colorimetry, and Folin–Ciocalteu methods). However, for more precise and specific bio-functional evaluation of these berries, such as qualitative and quantitative analysis of specific anthocyanins, polyphenols, flavonoids, and other secondary metabolites, it is necessary to incorporate advanced instrumentation such as high-performance liquid chromatography (HPLC) or liquid chromatography–mass spectrometry (LC/MS) in future studies.

### 3.5. Correlation and Principal Component Analysis

To comprehensively understand the interrelationship between the fruit quality traits, correlation analysis was conducted on the 17 fruit quality traits of each cultivar (Figure 2, Appendix A). Generally, for all the three cultivars, Ri, Ff, 4 °C-Sl, 25 °C-Sl, and TA showed significant positive correlations with each other and showed significant negative correlations with SS:TA. For ‘Lanjingling’ (Figure 2A, Appendix A), SS:TA showed a significant positive correlation with *L** (*r* = 0.57) and vitamin C (0.59) while showing a significant negative correlation with Ff (*r* = −0.88), Ri (−0.81), 25 °C-SL (−0.74), 4 °C-Sl (−0.78), and TA (−0.95), respectively. Furthermore, vitamin C showed a significantly negative correlation with Ri (−0.57), 4 °C-Sl (−0.53), SS (−0.52), and TA (−0.53), respectively. However, no significant correlation was found between phenolics or anthocyanins and any traits. For ‘Wulan’ (Figure 2B, Appendix A), SS:TA exhibited a significant positive correlation with fruit size (0.52) and SS (0.65), respectively, while showing a negative correlation with Ff (−0.80), Ri (−0.70), 25 °CS-Sl (−0.66),4 °C-SL (−0.76), TA (−0.86), and phenolic content (−0.66), respectively. Meanwhile, the phenolics exhibited a significant positive correlation with TA (0.65), Ri (0.78), 25 °C-SL (0.62), 4 °C-SL (0.84), and Ff (0.83), respectively, while exhibiting a significant negative correlation with fruit length (−0.63), width (−0.69), size (−0.74), weight (−0.74), and SS:TA, respectively. The anthocyanins exhibited a significant positive correlation with *L** (0.80) while displaying a negative correlation with *a** and *b** (−0.71 and −0.77), respectively. However, no significant correlation was observed between vitamin C and any other traits. For ‘Berel’ (Figure 2C, Appendix A), SS:TA showed a significant positive correlation with *L** (0.56) while exhibiting a negative correlation with *a** (−0.63), *b** (−0.64), Ff (−0.80), Ri (−0.77), 25 °C-Sl (−0.72), 4 °C-SL (−0.72), TA (−0.52) and phenolics (−0.67) respectively. The phenolic content showed a significant positive correlation with *a** (0.83), *b** (0.59), Ff (0.61), Ri (0.61), 25 °C-Sl (0.65), 4 °C-SL (0.77), respectively; while exhibited a significant negative correlation with *L** (−0.75), vitamin C (−0.56) and SS:TA. A significant positive correlation was found between the anthocyanins and vitamin C (0.73). The vitamin C showed a negative correlation with 4°C-SL (−0.56) and phenolics (−0.56).

To gain a deeper understanding of the interrelationship between the key quality traits and harvesting dates for each cultivar, principal component analysis (PCA) was performed based on 11 traits (weight, size, SS, TA, SS:TA, Ff, 4 °C-Sl, 25 °C-SL, anthocyanins, vitamin C, and phenolics). Taking eigenvalue greater than 1.0 into account, three, three, and four principal components (PC) were extracted in the ‘Lanjingling’, ‘Wulan’, and ‘Berel’, respectively (Appendix A), and thus 3D PCA was plotted (Figure 3). For ‘Lanjingling’ (Figure 3A), the three components account for 75.67% of the total variance; the first, second, and third PC (PC1, PC2, and PC3) contributes to 52.08%, 13.72%, and 9.86% of the total variance, respectively; in the 3D coordinate, the loadings of Ff, 25 °C-Sl, 4 °C-Sl, TA, and SS are clearly associated with each other, and the observations of 62DAF were relatively well clustered. For ‘Wulan’ (Figure 3B), PC1 (53.87%), PC2 (14.47%), and PC3 (10.45%) collectively accounted for 78.79% of the total variance; specifically, the loadings of Ff, 25 °C-Sl, 4 °C-Sl, and TA are clearly associated with each other, and the observations of 58DAF and 62DAF were respectively well clustered in the coordinate. For ‘Berel’ (Figure 3C), PC1 (44.99%), PC2 (17.56%), and PC3 (14.60%) collectively account for 77.15% of the total variance; the loadings of Ff, 25 °C-Sl, and 4 °C-Sl are clearly associated with each other, and the observations of 62DAF and 67DAF were respectively well clustered.

The SS:TA, fruit firmness, shelf life, and health-promoting phytochemicals are crucial fruit quality traits for blue honeysuckle berries, which majorly determine the industry expansion for berry crops [53,54]. The SS:TA ratio plays a crucial role in determining the primary taste and flavor of fruits [54,55]. Fruit firmness directly influences fruit quality preference, transportability, shelf life, and the feasibility of mechanical harvesting for small berry crops [56,57]. Extending shelf life is essential to enhancing the profitability and commercial availability of fruits while maintaining optimal quality [58], thus attracting significant attention from berry growers, dealers, and breeders. In the present study, significant positive correlations were observed between Ri, Ff, 25°C-Sl, 4°C-Sl, and TA; additionally, significant negative correlations were found between SS:TA and each of the aforementioned five traits. The sustainable increasing pattern of SS:TA along with delayed harvest (Table 3) and the feasibility of measuring SS:TA compared to Ff, Ri, and Sl suggests that SS:TA is not only a crucial index in fruit sensory evaluation but also serves as an ideal indicator for monitoring the fruit maturity and determining harvest timing for blue honeysuckle. However, limited correlation relationships were found between the three phytochemicals and the other traits (Figure 2 and Figure 3) added to the irregular accumulation trends of the three phytochemicals during the harvest windows (Table 4), suggesting that the accumulations of anthocyanins, vitamin C, and phenolics in blue honeysuckle berries are independent of the other measured traits in the present study and additional data obtained through various technological approaches should be included to better understand the accumulation pattern of these compounds during fruit ripening in the future.

### 3.6. Evaluation of Harvest Dates Based on Factor Analysis

To provide practical and direct information on blue honeysuckle harvesting for producers and researchers, we condensed the 17 fruit quality traits into nine key commercial indicators (weight, size, SS:TA ratio, Ff, 4 °C-Sl, 25 °C-Sl, anthocyanins, vitamin C, and phenolics). Additionally, factor analysis was employed to comprehensively evaluate the harvest dates for each cultivar. The KMO statistics for the datasets of ‘Lanjingling’, ‘Wulan’, and ‘Berel’ were 0.633, 6.682, and 0.668, respectively (Appendix A). The Bartlett’s spherical test chi-square statistics were 80.216, 106.411, and 79.467 for the respective datasets as well, with all significance levels being less than 0.05. These results indicate that the datasets were suitable for factor analysis. Using an eigenvalue criterion greater than 1.00, three factors were respectively extracted from the datasets of ‘Lanjingling’, ‘Wulan’, and ‘Berel’, with cumulative contribution rates of 75.22%, 83.92%, and 82.78%, respectively (Appendix A). To enhance the interpretability of the factor variables, variance maximization orthogonal rotation was conducted on the factor matrixes. As depicted in Appendix A, for ‘Lanjingling’, the first factor (FC1) exhibited strong associations with SS:TA, firmness, 4 °C-Sl, and 25 °C-Sl; the second factor (FC2) displayed strong associations with weight, size and phenolics; the third factor (FC3) showed strong associations with vitamin C and anthocyanins. For ‘Wulan’, the FC1 showed strong associations with firmness, 4 °C-Sl, 25 °C-Sl, and SS:TA; the FC2 displayed strong associations with weight, phenolics, and size; the FC3 exhibited strong associations with vitamin C and anthocyanins. For ‘Berel’, FC1 demonstrated strong associations with firmness, 25 °C-Sl, and SS:TA; FC2 showed strong associations with phenolics, vitamin C, and anthocyanins; FC3 exhibited strong associations with 4 °C-Sl, weight, and size. The comprehensive score for each observation was computed based on the FC loadings of each observation and the variance contribution rate of each FC (Appendix A). As summarized in Table 5, the comprehensive scores for ‘Lanjingling’, ‘Wulan’, and ‘Berel’ across the five harvests ranged from −0.78 (67DAF) to 1.05 (47DAF), −0.63 (62DAF) to 0.46 (54DAF), and −0.67 (62DAF) to 0.82 (52DAF), respectively.

Based on the factor analysis results and in combination with the investigation of fruit traits (Table 1, Table 2, Table 3 and Table 4), our findings suggest that the suitable harvest date for ‘Lanjingling’ is either at 47DAF (SS:TA ≈ 10), yielding berries characterized by the longest shelf life, highest firmness, and highest levels of anthocyanins and phenolics; or at 67DAF (SS:TA ≈ 16.0), yielding berries with highest vitamin C content and sweetness, as well as largest fruit size and weight. Considering the potential cultivation of ‘Lanjingling’ as a table variety [4], these two distinct harvest dates might correspond, respectively, to long-distance fresh consumption and local fresh consumption. For ‘Wulan’, the results suggest the suitable harvest date is either at 54DAF (SS:TA ≈ 9.0), yielding berries characterized by the highest levels of anthocyanins and vitamin C, or at 62DAF (SS:TA > 10.0), yielding berries with the highest sweetness, largest fruit size, and weight. As ‘Wulan’ serves as a dual-purpose cultivar for both table consumption and processing purposes [4], these two distinct dates might correspond, respectively, to utilization in processing and local fresh markets. For ‘Berel’, the findings suggest the suitable harvest date is either at 52DAF (SS:TA ≈ 6.5), yielding berries characterized by the highest levels of anthocyanins and vitamin C, or at 62DAF (SS:TA > 7.0), yielding berries with above-average levels of sweetness, size, and weight. Considering its dominant purpose of processing, high acid levels, high yield, and anti-abscission traits [59,60,61], these results also imply the optimal harvest date for ‘Berel’ depends on the market demand.

## 4. Conclusions

In order to determine the appropriate harvest dates for three major cultivars of blue honeysuckle in China, this study investigated 17 fruit traits in their respective harvest windows. Overall, with the delay in harvesting, there were increasing trends in SS:TA and decreasing trends in firmness and shelf life. Significant negative correlations were found between SS:TA and fruit firmness and respiration intensity, as well as shelf life, indicating that SS:TA serves as an ideal indicator for monitoring the maturity of blue honeysuckle berries. Comprehensively, the most suitable harvest dates for ‘Lanjingling’ were 47DAF (SS:TA ≈ 10.0) and 67DAF (SS:TA ≈ 16.0), for ‘Wulan’ were 54DAF (SS:TA ≈ 9) and 62DAF (SS:TA > 10.0), and for ‘Berel’ were 52DAF (SS:TA ≈ 6.5) and 62DAF (SS:TA > 7.0).

## Figures and Tables

**Figure 1 plants-12-03758-f001:**
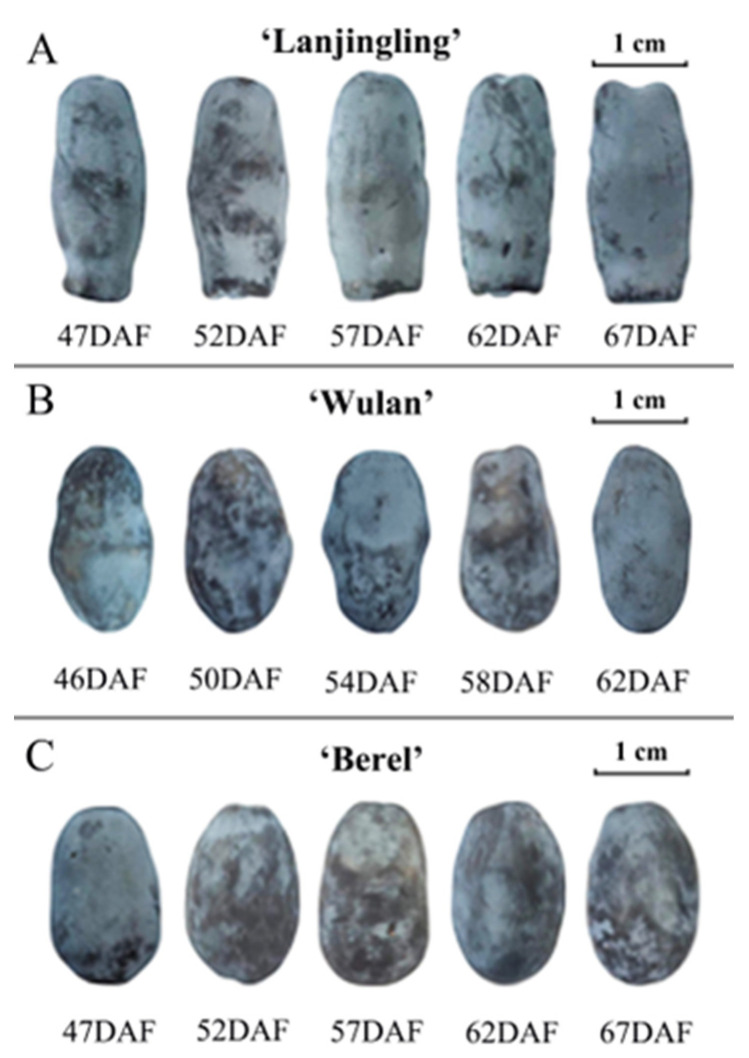
Fruit appearance of the three blue honeysuckle cultivars in their respective harvest windows. (**A**) ‘Lanjingling’; (**B**) ‘Wulan’; (**C**) ‘Berel’.

**Figure 2 plants-12-03758-f002:**
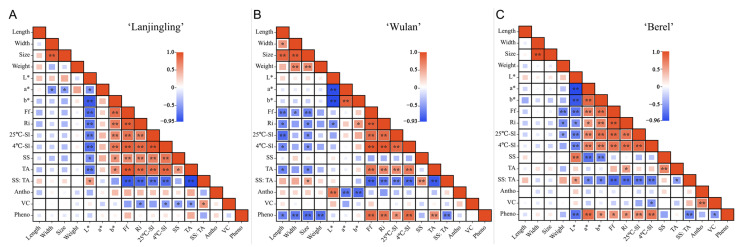
Correlation analysis of fruit quality traits in ‘Lanjingling’, ‘Wulan’, and ‘Berel’ during their respective harvest windows. (**A**) ‘Lanjingling’; (**B**) ‘Wulan’; (**C**) ‘Berel’. Abbreviations: SS (soluble solids), TA (acidity), Vc (vitamin C), Pheno (phenolic compounds), Antho (anthocyanins), Sl (shelf life), Ff (fruit firmness), and Ri (respiration intensity). * and ** represent statistical significance at *p* ≤ 0.05 and *p* ≤ 0.01 levels, respectively.

**Figure 3 plants-12-03758-f003:**
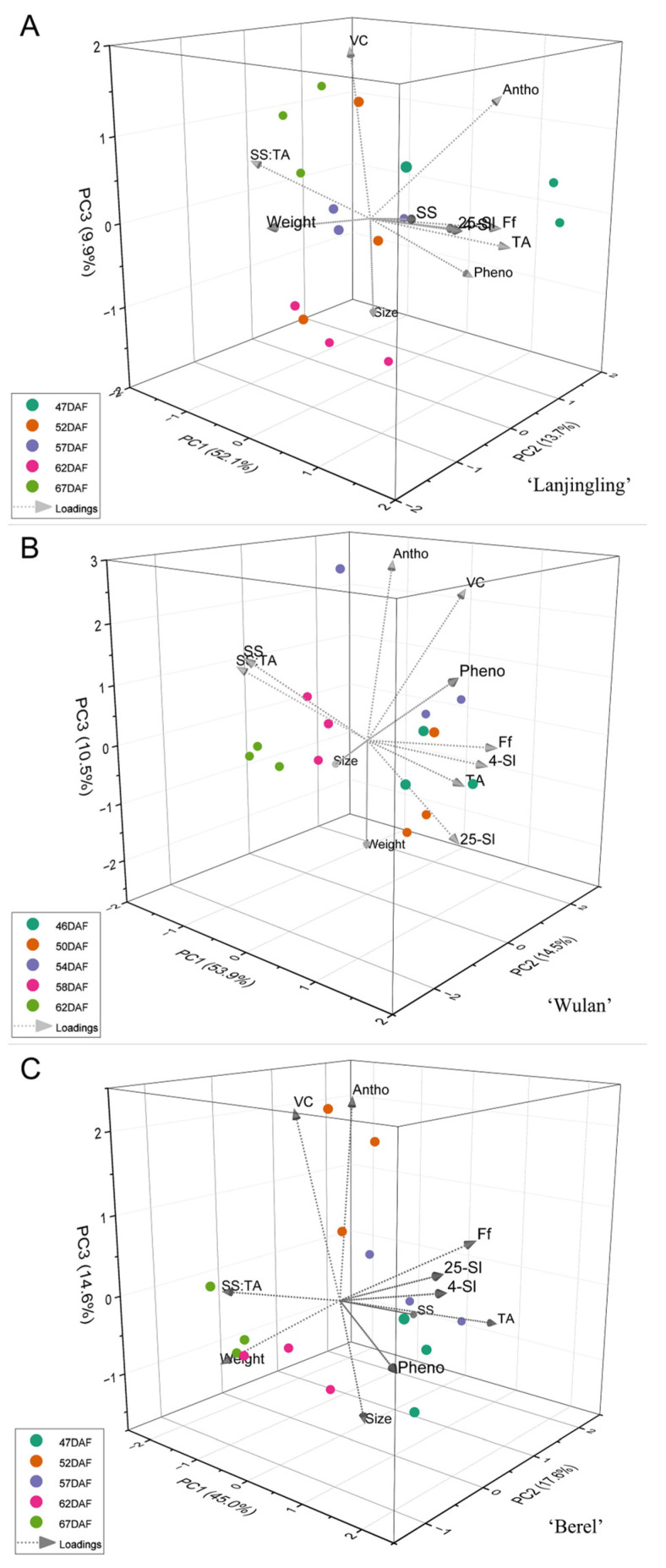
Principal component bi-plot of the 5 harvest dates for the three blue honeysuckle cultivars (**A**) ‘Lanjingling’; (**B**) ‘Wulan’; (**C**) ‘Berel’.

**Table 1 plants-12-03758-t001:** Fruit length, width, size, weight and color of the three blue honeysuckle cultivars in their respective harvest windows.

Cultivar	Harvest Time	Length (mm)	Width (mm)	Size (cm^3^)	Weight (g)	*L**	*a**	*b**
Lanjingling	47DAF	23.08 ± 0.86 a	9.50 ± 0.24 a	1.64 ± 0.11 a	1.18 ± 0.09 a	33.99 ± 0.98 b	−1.36 ± 0.18 a	−5.21 ± 0.24 a
	52DAF	23.45 ± 0.15 a	9.56 ± 0.27 a	1.68 ± 0.10 a	1.19 ± 0.04 a	36.19 ± 1.35 ab	−1.38 ± 0.05 a	−5.45 ± 0.26 a
	57DAF	23.57 ± 0.84 a	9.61 ± 0.34 a	1.71 ± 0.08 a	1.17 ± 0.11 a	38.81 ± 1.77 a	−1.45 ± 0.37 a	−6.66 ± 0.27 b
	62DAF	23.78 ± 0.34 a	9.61 ± 0.27 a	1.73 ± 0.08 a	1.19 ± 0.03 a	39.45 ± 0.13 a	−1.47 ± 0.06 a	−6.77 ± 0.17 b
	67DAF	23.63 ± 0.22 a	9.65 ± 0.23 a	1.73 ± 0.10 a	1.19 ± 0.06 a	38.40 ± 1.46 a	−1.48 ± 0.13 a	−6.36 ± 0.03 b
Wulan	46DAF	18.30 ± 0.57 b	10.37 ± 0.20 a	2.06 ± 0.13 b	1.02 ± 0.02 a	31.98 ± 0.83 b	−1.36 ± 0.11 ab	−5.58 ± 0.55 b
	50DAF	19.81 ± 0.05 a	10.54 ± 0.43 a	2.31 ± 0.19 ab	1.04 ± 0.06 a	28.77 ± 0.73 c	−1.04 ± 0.20 a	−4.23 ± 0.47 a
	54DAF	20.00 ± 0.47 a	10.65 ± 0.14 a	2.37 ± 0.09 ab	1.06 ± 0.04 a	37.24 ± 0.28 a	−1.84 ± 0.07 c	−6.80 ± 0.32 c
	58DAF	20.03 ± 0.29 a	10.73 ± 0.17 a	2.42 ± 0.12 a	1.06 ± 0.02 a	36.51 ± 1.18 a	−1.74 ± 0.16 bc	−7.24 ± 0.50 c
	62DAF	20.06 ± 0.11 a	10.75 ± 0.08 a	2.42 ± 0.03 a	1.07 ± 0.02 a	31.55 ± 1.99 bc	−1.01 ± 0.13 a	−5.40 ± 0.38 ab
Berel	47DAF	17.32 ± 0.06 a	11.05 ± 0.12 a	2.21 ± 0.05 a	1.12 ± 0.01 a	28.26 ± 1.12 c	−0.91 ± 0.10 a	−3.65 ± 0.23 a
	52DAF	17.35 ± 0.11 a	10.99 ± 0.29 a	2.20 ± 0.11 a	1.12 ± 0.03 a	30.24 ± 0.91 c	−1.23 ± 0.09 b	−3.99 ± 0.21 a
	57DAF	17.37 ± 0.17 a	11.05 ± 0.25 a	2.22 ± 0.11 a	1.13 ± 0.03 a	34.58 ± 0.79 a	−1.69 ± 0.12 d	−5.24 ± 0.18 b
	62DAF	17.38 ± 0.06 a	11.05 ± 0.35 a	2.23 ± 0.14 a	1.14 ± 0.03 a	32.31 ± 0.17 b	−1.52 ± 0.07 cd	−4.84 ± 0.34 b
	67DAF	17.40 ± 0.14 a	11.10 ± 0.26 a	2.25 ± 0.08 a	1.15 ± 0.01 a	32.55 ± 0.35 b	−1.41 ± 0.01 bc	−5.06 ± 0.33 b

Values are presented as mean ± standard deviation. Mean values denoted by different letters (a–d) within the same row indicate statistically significant differences at *p* < 0.05. *L** is the lightness value, which defines black at 0 and white at 100; *a** represents the green-magenta opponents (negative values toward green and positive toward magenta); *b** represents the blue-yellow opponents (negative values toward blue and positive toward yellow).

**Table 2 plants-12-03758-t002:** Fruit firmness, respiration intensity, and shelf life of the three blue honeysuckle cultivars in their respective harvest windows.

Cultivar	Harvest Time	Firmness (N)	Respiration Intensity (CO_2_, mg/kg·h)	Shelf Life under 25 °C (Day)	Shelf Life under 4 °C (Day)
Lanjingling	47DAF	3.87 ± 0.13 a	41.33 ± 1.30 a	12.33 ± 2.52 a	26.33 ± 3.51 a
	52DAF	2.92 ± 0.15 b	40.52 ± 3.95 a	9.00 ± 2.65 ab	21.33 ± 2.52 ab
	57DAF	2.21 ± 0.04 c	27.94 ± 1.18 b	8.33 ± 2.52 ab	17.67 ± 3.06 bc
	62DAF	2.14 ± 0.12 c	22.53 ± 2.23 b	6.00 ± 2.65 ab	14.67 ± 1.53 bc
	67DAF	1.44 ± 0.06 d	12.43 ± 1.95 c	4.33 ± 1.53 b	11.00 ± 3.61 c
Wulan	46DAF	2.23 ± 0.06 a	80.74 ± 2.87 a	11.33 ± 3.21 a	21.00 ± 1.00 a
	50DAF	1.92 ± 0.11 b	80.30 ± 2.65 a	8.00 ± 3.61 ab	19.00 ± 2.00 a
	54DAF	1.68 ± 0.15 b	57.56 ± 2.23 b	4.33 ± 1.53 b	13.67 ± 1.15 b
	58DAF	1.18 ± 0.07 c	49.30 ± 4.03 c	3.00 ± 0.00 b	10.67 ± 1.53 b
	62DAF	0.88 ± 0.03 d	47.96 ± 2.55 c	2.33 ± 0.58 b	5.00 ± 1.00 c
Berel	47DAF	2.94 ± 0.02 a	48.46 ± 1.80 a	12.00 ± 3.00 a	24.67 ± 2.08 a
	52DAF	2.52 ± 0.03 b	40.60 ± 3.17 b	7.00 ± 1.73 ab	16.00 ± 1.73 b
	57DAF	2.01 ± 0.09 c	37.83 ± 2.31 b	4.00 ± 1.73 b	15.67 ± 2.08 b
	62DAF	1.90 ± 0.17 c	26.06 ± 1.02 c	4.33 ± 1.53 b	12.67 ± 2.08 b
	67DAF	1.34 ± 0.02 d	22.36 ± 2.67 c	2.33 ± 0.58 b	12.33 ± 2.31 b

Values are presented as mean ± standard deviation. Mean values denoted by different letters (a–d) within the same row indicate statistically significant differences at *p* < 0.05.

**Table 3 plants-12-03758-t003:** Soluble solids, total acid and solid-acid ratio of the three blue honeysuckle cultivars in their respective harvest windows.

Cultivar	Harvest Time	Soluble Solids (SS, %)	Titratable Acid (TA, %)	SS:TA
Lanjingling	47DAF	15.15 ± 0.47 a	1.52 ± 0.08 a	10.00 ± 0.73 c
	52DAF	15.10 ± 0.29 a	1.18 ± 0.01 b	12.83 ± 0.33 b
	57DAF	14.15 ± 0.31 a	1.14 ± 0.08 b	12.44 ± 0.73 b
	62DAF	14.44 ± 0.97 a	1.09 ± 0.09 b	13.28 ± 1.47 b
	67DAF	13.84 ± 0.37 a	0.86 ± 0.02 c	16.01 ± 0.78 a
	Average	14.54 ± 0.58	1.16 ± 0.23	12.91 ± 2.15
Wulan	46DAF	17.23 ± 0.88 a	2.12 ± 0.06 a	8.13 ± 0.22 b
	50DAF	16.13 ± 0.76 a	1.97 ± 0.09 ab	8.20 ± 0.46 b
	54DAF	16.90 ± 1.68 a	1.91 ± 0.12 bc	8.89 ± 1.33 ab
	58DAF	17.41 ± 0.51 a	1.85 ± 0.15 bc	9.44 ± 0.63 ab
	62DAF	17.65 ± 0.58 a	1.73 ± 0.12 c	10.23 ± 0.61 a
	Average	17.06 ± 0.59	1.92 ± 0.14	8.98 ± 0.88
Berel	47DAF	12.88 ± 0.96 b	2.23 ± 0.08 b	5.77 ± 0.28 b
	52DAF	13.52 ± 0.76 b	2.06 ± 0.07 bc	6.57 ± 0.39 ab
	57DAF	16.48 ± 0.08 a	2.51 ± 0.13 a	6.59 ± 0.31 ab
	62DAF	14.31 ± 0.32 b	2.02 ± 0.09 bc	7.08 ± 0.24 a
	67DAF	13.24 ± 0.13 b	1.86 ± 0.09 c	7.11 ± 0.43 a
	Average	14.09 ± 1.44	2.14 ± 0.24	6.62 ± 0.54

Values are presented as mean ± standard deviation. Mean values denoted by different letters (a–c) within the same row indicate statistically significant differences at *p* < 0.05.

**Table 4 plants-12-03758-t004:** Key phytochemical contents of the three blue honeysuckle cultivars during their respective harvest windows.

Cultivars	Harvest Time	Anthocyanins (mg/100 g)	Vitamin C (mg/100 g)	Phenolics (mg/g)
Lanjingling	47DAF	276.83 ± 20.07 a	91.76 ± 0.77 bc	37.59 ± 3.83 a
	52DAF	250.29 ± 26.55 a	90.22 ± 0.77 bc	25.22 ± 0.85 b
	57DAF	253.41 ± 12.87 a	92.52 ± 1.54 b	27.58 ± 0.94 b
	62DAF	233.85 ± 17.92 a	88.44 ± 1.77 c	35.57 ± 1.53 a
	67DAF	254.08 ± 11.38 a	99.68 ± 2.21 a	28.88 ± 2.14 b
	Average	253.69 ± 15.34	92.52 ± 4.30	30.97 ± 5.34
Wulan	46DAF	255.34 ± 34.41 bc	121.16 ± 2.21 c	36.52 ± 0.06 a
	50DAF	236.38 ± 7.90 c	137.27 ± 4.50 b	33.41 ± 3.72 ab
	54DAF	312.23 ± 21.56 a	191.23 ± 6.82 a	30.55 ± 1.78 abc
	58DAF	305.91 ± 14.36 ab	108.13 ± 0.89 d	29.79 ± 3.64 bc
	62DAF	256.91 ± 2.19 bc	116.82 ± 2.69 cd	25.40 ± 1.05 c
	Average	273.29 ± 33.70	134.92 ± 33.21	31.14 ± 4.16
Berel	47DAF	240.17 ± 24.09 b	89.71 ± 0.88 c	50.00 ± 2.78 a
	52DAF	334.98 ± 9.54 a	120.40 ± 7.13 a	39.92 ± 0.62 b
	57DAF	308.43 ± 22.85 a	103.78 ± 1.93 b	38.74 ± 0.38 b
	62DAF	235.71 ± 13.68 b	101.48 ± 1.17 b	37.66 ± 2.18 b
	67DAF	290.74 ± 17.10 a	107.10 ± 2.77 b	41.28 ± 2.05 b
	Average	282.01 ± 43.23	104.49 ± 11.04	41.52 ± 4.93

Values are presented as mean ± standard deviation. Mean values denoted by different letters (a–d) within the same row indicate statistically significant differences at *p* < 0.05.

**Table 5 plants-12-03758-t005:** Factor scores of the three cultivars’ five harvest dates.

Cultivar	Harvest Time	FC1 Score	FC2 Score	FC3 Score	Comprehensive Score
Lanjingling	47DAF	1.37	1.02	0.04	1.05
	52DAF	0.28	−0.35	0.48	0.20
	57DAF	−0.09	−0.08	−0.01	−0.08
	62DAF	−0.03	−1.23	−0.72	−0.39
	67DAF	−1.53	0.65	0.20	−0.78
Wulan	46DAF	1.00	−0.97	−0.56	0.13
	50DAF	0.84	0.11	−0.47	0.38
	54DAF	0.12	0.22	1.81	0.46
	58DAF	−0.74	0.14	−0.01	−0.34
	62DAF	−1.22	0.50	−0.77	−0.63
Berel	47DAF	1.44	−1.20	0.26	0.49
	52DAF	0.62	1.57	0.10	0.82
	57DAF	−0.16	0.35	−0.04	0.00
	62DAF	−0.85	−0.62	−0.10	−0.67
	67DAF	−1.05	−0.10	−0.22	−0.65

## Data Availability

The data supporting the results in the present work is shown in Appendix A.

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
