# Peer review of "Evaluation of the Harvest Dates for Three Major Cultivars of Blue Honeysuckle (Lonicera caerulea L.) in China"

_plants, 2023, doi:10.3390/plants12213758_

Round 1

Reviewer 1 Report

Comments and Suggestions for Authors

The author of the manuscript need to focus on the following points

Include the ethnobotanical, traditional, industrial values of Lonicera caerulea in the introductory section.

 Include the phytochemical analysis of Lonicera caerulea cultivars using HPLC, LC-MS/MS.

Discussion is very poor. Strengthen your results by using more recent citations in similar fileds.

Avoid using data (numerical values) in the conclusion section, as these data are already methined in the results sections.

Author can perform and submit the total flavonoid contents data of three different cultivars.

Reviewer 2 Report

Comments and Suggestions for Authors

The english can be improved along the text, either way the presentation is very good and results are robust. It would improve an increase in bibliographi references.

Comments on the Quality of English Language

English is good but can be improved.

Reviewer 3 Report

Comments and Suggestions for Authors

The manuscript ‘Optimal harvest timing assessment for three main blue honey-suckle berry (Lonicera caerulea L.) cultivars in China’ needs minor revision.

Add the novelty of this work at the end of the introduction.

Italicize the color parameters L a b in the whole text.

Change TSS unit to %

improve discussion

Correct reference format based on journal requirements.

Reviewer 4 Report

Comments and Suggestions for Authors

The paper is an interesting issue that the authors deal with. For the best possible quality, finding a good time to pick the fruits is necessary. I believe the methods of isolating chemical compounds and their determination should have been described in more detail.

Round 2

Reviewer 1 Report

Comments and Suggestions for Authors

The author of the manuscript has revised it with proper answers and corrected the  manuscript and responded to all the comments.

Now the manuscript can be accepted inn ites present form.

Author Response

Many thanks to you for making our research better.